# Single molecule mechanics resolves the earliest events in force generation by cardiac myosin

**Michael S Woody[1], Donald A Winkelmann[2], Marco Capitanio[3,4], E Michael Ostap[5]\*, Yale E Goldman[5]\***

[1]Graduate Group in Biochemistry and Molecular Biophysics, Perelman School of Medicine, University of Pennsylvania, Philadelphia, United States; [2]Department of Pathology and Laboratory Medicine, Robert Wood Johnson Medical School, Rutgers University, Piscataway, United States; [3]LENS - European Laboratory for Non-linear Spectroscopy, Sesto Fiorentino, Italy; [4]Department of Physics and Astronomy, University of Florence, Sesto Fiorentino, Italy; [5]Pennsylvania Muscle Institute, Perelman School of Medicine, University of Pennsylvania, Philadelphia, United States

**Abstract** Key steps of cardiac mechanochemistry, including the force-generating working stroke and the release of phosphate ($P_i$), occur rapidly after myosin-actin attachment. An ultra-high-speed optical trap enabled direct observation of the timing and amplitude of the working stroke, which can occur within <200 μs of actin binding by β-cardiac myosin. The initial actomyosin state can sustain loads of at least 4.5 pN and proceeds directly to the stroke or detaches before releasing ATP hydrolysis products. The rates of these processes depend on the force. The time between binding and stroke is unaffected by 10 mM $P_i$ which, along with other findings, indicates the stroke precedes phosphate release. After $P_i$ release, $P_i$ can rebind enabling reversal of the working stroke. Detecting these rapid events under physiological loads provides definitive indication of the dynamics by which actomyosin converts biochemical energy into mechanical work.
DOI: https://doi.org/10.7554/eLife.49266.001

**\*For correspondence:**
ostap@pennmedicine.upenn.edu (EMO);
goldmany@upenn.edu (YEG)

**Competing interests:** The authors declare that no competing interests exist.

## Introduction

Myosin is a cytoskeletal motor that uses metabolic energy stored in ATP to do mechanical work. This process underlies muscle contraction, cell motility, and intracellular transport. In muscle, myosin causes displacements of actin filaments connected to work against an external load. The majority of the mechanical work done by myosin is performed in the working stroke, the rotation of the myosin lever arm domain, which occurs soon after actin binding and is closely associated with the release of inorganic phosphate ($P_i$), a product of ATP hydrolysis. Understanding the connections between the formation of a force-bearing actin-bound state, the biochemical step of $P_i$ release, and the mechanical action of the working stroke is fundamental to understanding energy transduction. Although substantial physiological and biophysical experiments have provided key details associated with these transitions (*Bagshaw and Trentham, 1974*; *Hibberd and Trentham, 1986*; *Hibberd et al., 1985*; *Cooke et al., 1988*; *Dantzig et al., 1992*; *Araujo and Walker, 1996*; *White et al., 1997*; *Takagi et al., 2006*; *Muretta et al., 2015*; *Caremani et al., 2015*; *Rohde et al., 2017*; *Llinas et al., 2015*), controversy remains regarding the order of the force-generating conformational transitions and whether the free energy change from phosphate release precedes or follows the working stroke (*Figure 1a*). Strengthening of the actomyosin bond due to conformational changes in myosin and actin, tilting of the lever arm, and $P_i$ release all occur within milliseconds of initial actin binding,

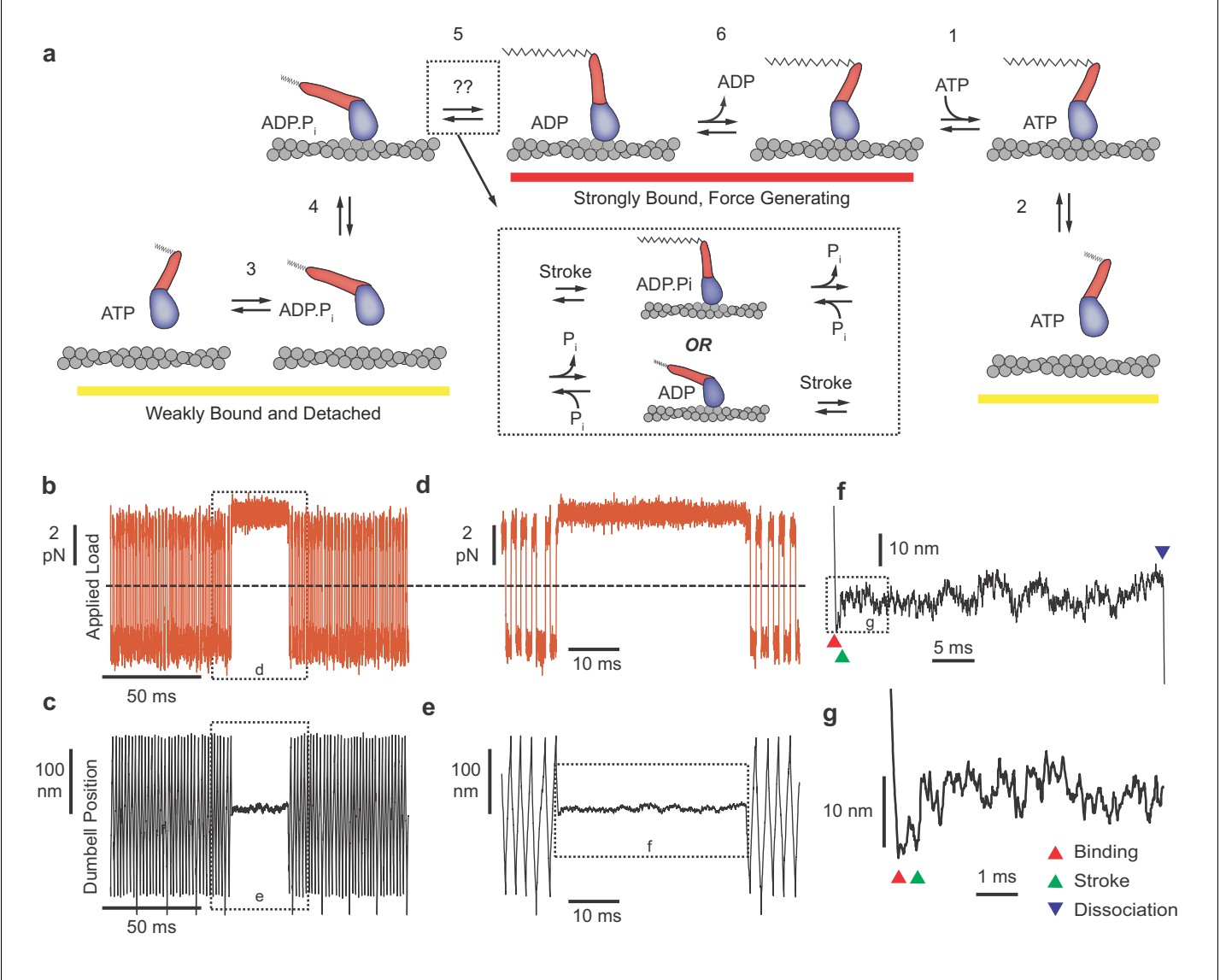

**Figure 1.** Myosin biochemical model and example UFFC traces. (a) Mechano-chemical model of acto-myosin interaction (*Goldman and Ostap, 2012*). Steps are numbered to remain consistent with the myosin literature. The working stroke is shown in two phases, the latter one coupled to ADP release from AM·ADP. Inset: The primary working stroke is depicted either before or after phosphate release from AM·ADP·P$_i$. (b) Applied forces acting on the bead-actin-bead dumbbell in the ultra-fast force clamp at ± 3.75 pN. Dashed line indicates zero force. (c) Position of the optical trap, which approximates the actin filament position. Dashed boxes in (b and c) indicate regions containing myosin binding events expanded in panels (d and e). (d and e) The dumbbell force and position signals expanded during a myosin binding event under hindering (positive) 3.75 pN load. (f) Further expanded section of panel f showing myosin binding (red triangle) and dissociation (blue triangle) distinguished by the sudden changes in slope of the position trace. (g) Further expanded view of (f) showing a displacement (green triangle) of ~10 nm occurring ~1 ms of actin binding.
DOI: https://doi.org/10.7554/eLife.49266.002

The following source data is available for figure 1:

**Source data 1.** Matlab figure files with data from *Figure 1*.
DOI: https://doi.org/10.7554/eLife.49266.003

making the mechanochemical steps difficult to resolve by commonly available biophysical measurements.

To detect the rapid sequence of events between actin binding and the working stroke, we examined the interactions of recombinant human beta-cardiac myosin (MHY7) with actin using the three-bead optical trapping geometry, where a bead-actin-bead 'dumbbell' assembly is brought near to a single myosin molecule attached to a pedestal bead (*Finer et al., 1994*). The myosin protein was a

heavy meromyosin (HMM) construct with a Flag-tag placed at the C-terminus for purification. It was attached non-specifically to nitrocellulose coated silica beads on the coverslip surface. The beta-cardiac isoform not only has great relevance as the dominant myosin in the human ventricles, but its slower kinetic properties compared to fast skeletal myosin facilitated detection of fast rate processes. Although this is a two-headed construct, we did not observe evidence that both heads interacted simultaneously with actin in the presence of ATP (see Discussion). We employed an ultra-fast force clamp (UFFC) which can resolve the onset of actin-myosin binding events to less than 100 microseconds and filament motions at sub-nm resolution (*Capitanio et al., 2012*). Unlike standard optical trapping techniques which typically have temporal resolutions > 10 ms, UFFC allows observation of the initial myosin working stroke which can occur within 1 ms of actin binding. We applied this system to explore actomyosin attachment durations and displacements in the presence of 1.5–4.5 pN mechanical loads in directions along the long-axis of the actin filament that hinder or assist the stroke. The duration of the events, and the timing and size of filament displacements during interactions were examined in the presence and absence of 10 mM $P_i$ to determine the relationship between actin binding, the working stroke, and $P_i$ release.

## Results

### Duration of actomyosin interactions observed under load

UFFC uses a feedback loop to maintain a constant force on an actin filament dumbbell by rapid (µs) control of the trapping laser position. The applied force (the geometric sum of the optical trap forces on each bead) causes the filament and two beads to move through the assay solution at a constant velocity set by the bead size and water viscosity and the force is alternated in direction to maintain the dumbbell within 200 nm excursion (*Figure 1b–e*). When a surface-attached myosin binds to actin and becomes loaded with the applied force, the filament motion stops within ~50 µs (*Figure 1d–g*). Displacement of the actin filament caused by the myosin working stroke is monitored during the interaction, which ends when myosin dissociates, allowing the actin filament to resume motion under the applied force (*Figure 1f*). The beginning and end of each actomyosin interaction are detected by examining the velocity of the optical traps as they respond to the feedback system, dropping to zero when myosin binds and increasing again when myosin releases the actin (*Figure 1f*). When the working stroke displaces the filament toward its pointed end, the beads and the traps move a corresponding amount to maintain a constant force (*Figure 1g*).

We first analyzed the duration of actomyosin interactions in the absence of $P_i$ under force that hinder or assist the stroke. To distinguish brief, reversible actomyosin attachments from longer attachments when myosins complete the ATPase cycle, most experiments were performed at a low (1 µM), rate-limiting MgATP concentration. Under both assisting and hindering loads, the distributions of event durations span a range from less than 1 ms to greater than 1 s (*Figure 2a–e*). The actomyosin events with durations less than 10 ms, which represent up to 50% of the observed interactions, have not been observed previously with standard optical trapping techniques because their durations are below the minimum detectable length (deadtime, 10 ms for standard techniques, 0.5–2.7 ms for the current technique) (*Capitanio et al., 2012*).

The very wide distributions of actomyosin event durations were best described by the linear combination of at least two exponential components, which we quantified using maximum likelihood estimation via the software package MEMLET (*Woody et al., 2016*). For forces > 2.25 pN, three exponential components were necessary to obtain an adequate fit (*Figure 2a and f*, *Figure 2—figure supplement 1*). This result contrasts with the distribution of durations observed with lower time-resolution, non-feedback measurements performed on the same myosin molecules, which are well-described by a single, slow rate of detachment (grey and black, *Figure 2a*).

The higher detachment rates, $k_f$ (*Figure 2f*, red closed symbols) and $k_{int}$ (green closed symbols), are more than 10-fold faster than the slowest detachment rate, $k_s$ (blue closed symbols). $k_f$ increases from $1 \times 10^3$ s$^{-1}$ at low loads to up to $3–6 \times 10^3$ s$^{-1}$ with increasing hindering or assisting forces. Events which detach at $k_f$ composed 20–40% ($A_f$) of all observed interactions (Supplementary *Figure 1b,d*). The events which detach at the intermediate rate make up a small fraction of observed interactions ($A_{int}$ = 10–20%, *Figure 2—figure supplement 1*) and detach at a rate, $k_{int}$ = $10^2–10^3$ s$^{-1}$. This rate has an asymmetric force dependence about zero force, increasing from −3.75 pN to

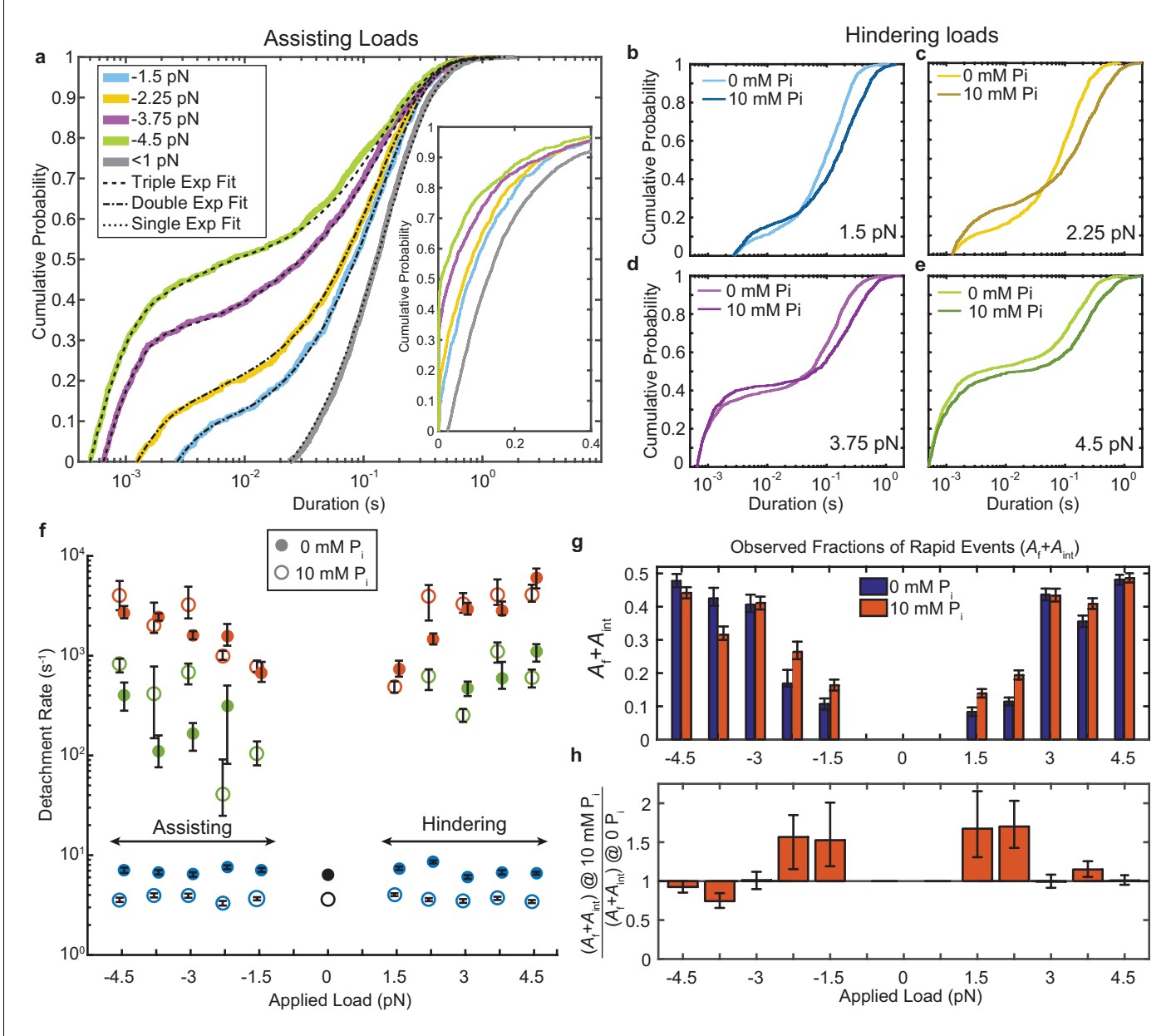

**Figure 2.** Kinetics of actomyosin attachment durations under mechanical load. (a) Cumulative distributions of actomyosin attachment event durations at 1 μM ATP under assisting loads (solid lines) plotted on semi-log axes. Best fits from the loglikelihood ratio test (*Woody et al., 2016*) are shown as dashed lines. Low-load data (<0.5 pN) from non-feedback experiments are shown in grey with a single exponential fit. The other fitted curves are double or triple exponential cumulative distributions as explained in the text. Inset shows the same data on a linear scale. (b–e) Comparison of cumulative distributions of durations under hindering loads listed in each panel with no added phosphate (lighter lines) and 10 mM added phosphate (dark lines). (f) Fitted rate constants from the multi-exponential functions which describe the data with no added phosphate (solid symbols) and 10 mM $P_i$ (open symbols). Positive forces (hindering loads) are defined as the direction that opposes the progression of the myosin working stroke. Blue symbols give the slowest of the fitted rates ($k_s$), red the fastest rates ($k_f$), and green the intermediate rate ($k_{int}$) in the cases where three rates were statistically justified. Black symbols are the single rate constant fitted to the non-feedback data from the same molecules. (g) Fraction of observed interactions that were rapid ($A_f + A_{int}$) from the multicomponent exponential fits for no added $P_i$ (blue) and 10 mM $P_i$ (red). (h) Relative fraction of rapid events for 10 mM added $P_i$ compared to 0 $P_i$, {($A_f+A_{int}$) at 10 mM $P_i$} / {($A_f+A_{int}$) at 0 mM $P_i$}. Error bars in f) – h) give 68% confidence intervals (90% in h)) from 500 rounds of bootstrapping (*Woody et al., 2016*). Numbers of events, 784 to 1502, from 5 to 8 molecules at each [$P_i$] (See *Supplementary file 1* Table 1 for details).

DOI: https://doi.org/10.7554/eLife.49266.004

The following source data and figure supplements are available for figure 2:

**Source data 1.** Matlab figure files with data from *Figure 2*.

*Figure 2 continued on next page*

*Figure 2 continued*

DOI: https://doi.org/10.7554/eLife.49266.008

**Figure supplement 1.** Attachment durations and fitted distributions.

DOI: https://doi.org/10.7554/eLife.49266.005

**Figure supplement 2.** Force dependence of fast and intermediate detachments at 0 $P_i$ and 1 μM MgATP.

DOI: https://doi.org/10.7554/eLife.49266.006

**Figure supplement 3.** Kinetics of actomyosin detachment in the presence of 1 mM MgATP.

DOI: https://doi.org/10.7554/eLife.49266.007

+4.5 pN (*Figure 2—figure supplement 2*). $k_s$, which is 6–8 $s^{-1}$ at 1 μM MgATP (consistent with the measured ATP binding rate) (*Liu et al., 2015*; *Deacon et al., 2012*; *Bloemink et al., 2014*), increases to 10–70 $s^{-1}$ in the presence of 1 mM MgATP (*Figure 2—figure supplement 3*). This observation is consistent with ADP release limiting the detachment rate at saturating ATP and indicates this population ($A_s$) represents motors that have completed the ATPase cycle (*Figure 1a*) (*Liu et al., 2015*; *Greenberg et al., 2014*; *Woody et al., 2018a*; *Sung et al., 2015*; *Liu et al., 2018*).

The actin attachments described by $k_f$ are much shorter than expected for molecules that undergo the full, multi-step biochemical reaction while attached to actin that ends with ATP-induced detachment (*Figure 1a*). These fast detaching interactions do not become faster when MgATP concentration is raised to 1 mM (*Figure 2—figure supplement 2*; *Figure 3*), further supporting the suggestion that they detach before completing the ATPase cycle. As shown below, these states do not undergo a working stroke, and are likely the short-lived states described in the physiological and biochemical literature as 'weak-binding' states (*Brenner, 1991*; *Stein et al., 1984*).

The fraction of rapid events detaching at $k_f$ or $k_{int}$ ($A_f + A_{int}$) increases with assisting and hindering forces (*Figure 2g* blue bars, *Figure 2h*). Although this effect may be partially due to reduced dead-times at higher forces (x-axis intercepts in *Figure 2a-e*, see Materials and methods), it also may indicate that higher forces are more likely to pull motor domains off of actin before they can complete their cycle (*Capitanio et al., 2012*). This hypothesis is further supported by plot of deadtime-corrected fraction $A_f + A_{int}$, which also increases with force (*Figure 2—figure supplement 1*).

## The myosin working stroke is rapid and more likely for long events

Individual myosin working stroke displacements were observed shortly after the linear motion of the actin stopped when myosin attached (*Figure 3a*). Upon binding, 5–10 nm displacements were observed in many traces. However, displacements were not readily apparent in all traces, and were rarely seen in interactions with durations below 5 ms (*Figure 3a*, bottom example). These short duration interactions likely represent myosins that detach before producing a working stroke (see below).

Attempts to objectively quantify these traces using step finding algorithms, hidden Markov chains, or Bayesian non-parametric analysis proved unsatisfactory due to the amplitude and frequency characteristics of the random signal fluctuations. Therefore, ensemble averaging of individual interactions was used to determine the average time and amplitude of the myosin working stroke at forces that hindered the stroke. Interactions were aligned at their detected start times and averaged forward in time (*Figure 3b,c*). Event displacements were extended in time from their last interacting positions to generate groups with equal durations that could be averaged (*Veigel et al., 1999*).

The sizes of the average displacements were highly force dependent, decreasing with increasing force (*Figure 3b,c*). At forces > 1.5 pN interactions with durations < 5 ms showed suppressed displacements (*Figure 3d*). When short events were excluded from the ensemble averages, the average displacement increased as the minimum event duration increased from 0 to 15 ms (*Figure 3e*, *Figure 3—figure supplement 1*). Average displacement of events > 15 ms decreased linearly with increasing hindering load (*Figure 3e–g*). If the load-dependence of displacement for interactions > 15 ms is due to compliance within the myosin, (*Dupuis et al., 1997*) then we estimate the stiffness of the motor to be 2.3 pN/nm (68% CI, 2.0–2.7 pN/nm) or greater from the linear fit to the data points in *Figure 3g*. This value is at the upper end of previous estimates (*Kaya and Higuchi, 2010*; *Lewalle et al., 2008*).

Strikingly, ensemble averages show that the myosin working stroke occurs within 5 ms of actin binding, without a detectable time lag at forces > 1.5 pN (*Figure 3f*, *Figure 4*). In *Figure 4a*, the traces are normalized to the amplitude of the initial stroke for visual comparison of their rates.

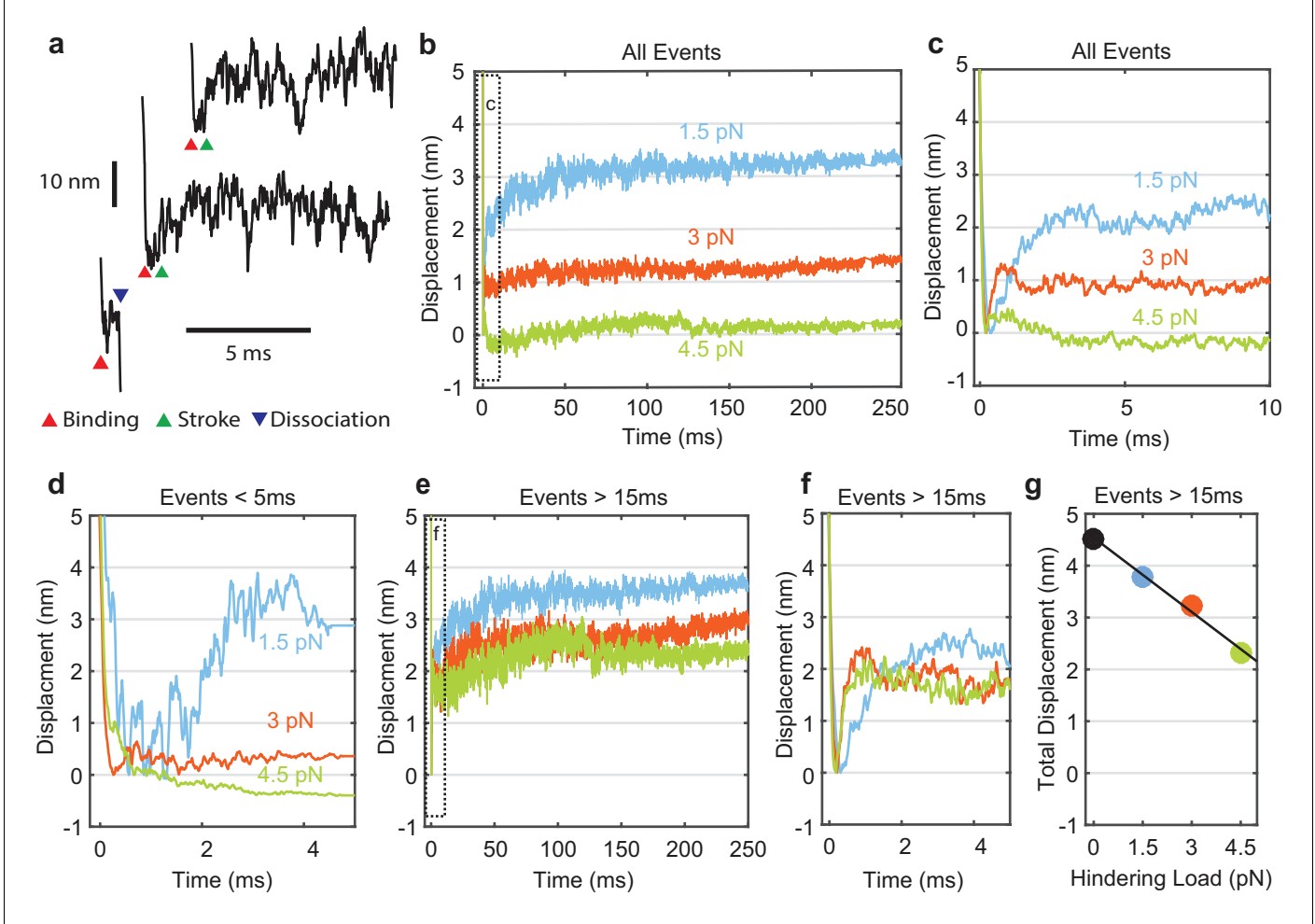

**Figure 3.** Myosin working stroke dynamics under hindering load. (a) Example traces of the position of the actin filament (reported by the leading trap position) during individual actomyosin interactions under 3.75 pN of hindering load. Red triangles indicate approximately when myosin binds to actin, stopping the motion of the filament. Green triangles indicate positive displacements expected from the myosin working stroke. The blue triangle shows dissociation for an event shorter than 5 ms. (b) Ensemble averages of all events from data at 0 added $P_i$ at 1.5, 3, and 4.5 pN hindering load. Dashed box shows a region that is shown expanded in panel (c). (d) Ensemble averages of events shorter than 5 ms. The average working stroke step is greatly reduced for forces above 1.5 pN apparently due to force-dependent detachment before the working stroke. (e) Ensemble averages of events lasting longer than 15 ms. (f) Expanded view of boxed area of (e) showing a similar initial displacement across forces. (g) Total measured step for events longer than 15 ms plotted against applied load with a linear fit, leading to an estimate of the stiffness of the myosin and its lever arm of at least 2.3 pN/nm.
DOI: https://doi.org/10.7554/eLife.49266.009

The following source data and figure supplement are available for figure 3:

**Source data 1.** Matlab figure files with data from *Figure 3*.
DOI: https://doi.org/10.7554/eLife.49266.011

**Figure supplement 1.** Quantification of ensemble average displacements from force-binned groups plotted against the minimum event duration (MED) included in each group for (a) 0 $P_i$ and (b) 10 mM added $P_i$.
DOI: https://doi.org/10.7554/eLife.49266.010

Estimates of the stroke rate from fits of the displacement data to single exponential rise (see Materials and methods) yielded rate constants ($k_{stroke}$) of 700 to 5,250 s$^{-1}$, that increase with force from 1.5 pN to 3 pN and then plateau (*Figure 4c*, blue). These observed rate constants are likely the sum of the rate for proceeding forward (performing the stroke, *Figure 1a*, $k_{+5}$) and rate for reversing actomyosin attachment (*Figure 1a*, $k_{-4}$, *Supplementary file 1* Table 2). In fact, only events in which the stroke is faster than myosin detachment are detected and thus can contribute to the stroke rate

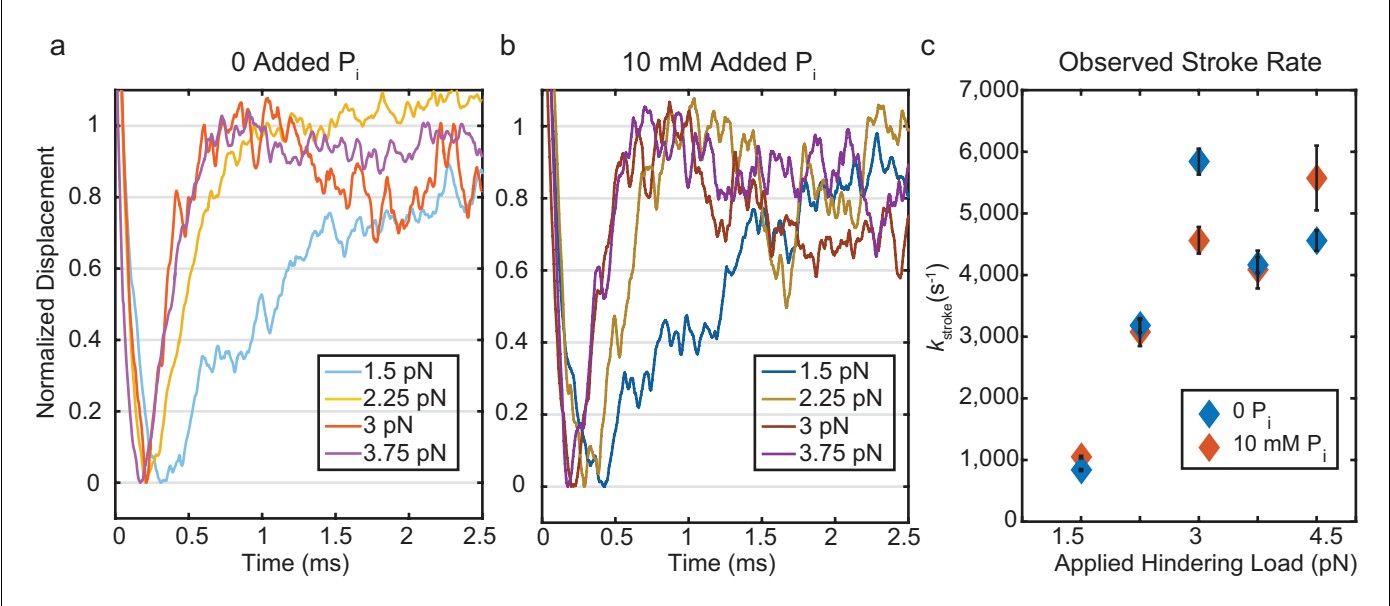

**Figure 4.** Kinetics of the initial working stroke displacement. (a) and (b) Ensembles averages from events longer than 25 ms plotted after normalizing the amplitudes of the initial displacements. (a) without added $P_i$ and (b) with 10 mM $P_i$ for 1.5 pN to 3.75 pN hindering loads. (c) Observed stroke rates from single exponential fits for no added $P_i$ (blue), and 10 mM added $P_i$ (red). Error bars show 95% confidence intervals from the fits.
DOI: https://doi.org/10.7554/eLife.49266.012
The following source data and figure supplement are available for figure 4:

**Source data 1.** Matlab figure files with data from *Figure 4*.
DOI: https://doi.org/10.7554/eLife.49266.014
**Figure supplement 1.** Models for the effect of free $P_i$ on single-molecule displacements in the presence of hindering mechanical load.
DOI: https://doi.org/10.7554/eLife.49266.013

in the ensemble average. Therefore, faster strokes are selected as the detachment rate $k_f$ increases with force.

A slower 1–1.5 nm further displacement (*Figure 3b,e*) observed in the ensemble averages was completed in 50–100 ms. The amplitude and time-scale of this slower displacement are consistent with the second phase of the myosin working stroke, previously associated with ADP release (*Greenberg et al., 2014*; *Woody et al., 2018a*; *Capitanio et al., 2006*).

## Effects of $P_i$ on the actomyosin attachments and displacement dynamics

The addition of phosphate to contracting demembranated muscle fibers results in a decrease in active force due to the redistribution of myosin states (*Hibberd et al., 1985*; *Cooke et al., 1988*; *Araujo and Walker, 1996*). To probe the effect of phosphate on myosin at the single molecule level, we performed UFFC experiments in the presence of 10 mM free $P_i$. Adding phosphate did not substantially change the rates of the $k_f$ or $k_{int}$ over the range of probed forces (*Figure 2f*, open red and green symbols); however, their combined amplitudes ($A_f+A_{int}$) increased in 10 mM $P_i$ at loads < 3 pN, suggesting an increase in the fraction of events that detach from weak-binding states (*Figure 2b,c,g,h*). The ratio of $A_f+A_{int}$ at 10 mM $P_i$ to $A_f+A_{int}$ at 0 $P_i$ shows there is up to a 50% increase in the number of these rapid detachments (*Figure 2h*). $k_s$ decreased about two-fold at 10 mM $P_i$ and 1 μM MgATP (*Figure 2f*) in both the UFFC and non-feedback experiments, which is consistent with weak competition between $P_i$ and ATP for binding at the active site, as previously proposed (*Amrute-Nayak et al., 2008*).

The presence of 10 mM free $P_i$ did not affect the rates of the ensemble displacements for events longer than 25 ms (*Figure 4b* and *Figure 4c* (red)). This result strongly implicates a mechanism in which $P_i$ release occurs after the working stroke, as a delay in the stroke kinetics would have been observed with 10 mM $P_i$ if $P_i$ release and rebinding occurred before the working stroke. (*Figure 4—*

*figure supplement 1*). Comparing the rate of the stroke to the fastest rate of detachment, $k_f$, from the event duration data (*Supplementary file 1* Table 2), shows that the average stroke rate is greater than or equal to the fastest phase of detachments. This result implies that molecules in the earliest state resolved in our experiment either perform a stroke or detach, and that $P_i$ release (or any other substantial intermediate step) does not occur between attachment and the stroke.

Many individual displacement records acquired in the presence of 10 mM $P_i$ showed reversals in the working stroke to the pre-stroke level (*Figure 5a*). However, the individual recordings are variable, and reversals cannot be reliably identified among the Brownian fluctuations. Thus, the dynamic features of the displacements were objectively quantified via ensemble averaging (*Figures 4* and *5 b-c*).

The average displacement traces decreased after the peak of the initial working stroke in the presence of 10 mM $P_i$ (*Figure 5c*, dark colors). We tabulated the total displacement, the size of any declines occurring after the initial positive displacement, and the time to reach the minimum of this dip (*Figure 5d–f*) (see Materials and methods for definition of the respective quantities). Total displacement was reduced (*Figure 5d*), and the size of the dip was increased by $P_i$ at all forces above 1.5 pN hindering load (*Figure 5e*). The changes in total displacement and the amplitude of dip are both greater at higher hindering loads, and the time to reach the minimum of this dip is considerably increased by the presence of 10 mM Pi (*Figure 5f*). These results are consistent with a model in which $P_i$ is released after the working stroke and then may rebind and cause reversal of the stroke (*Figure 4—figure supplement 1*). We performed simulations of individual actomyosin interactions using a kinetic model with these features (*Figure 5—figure supplement 1*). Ensemble averages of these interactions recapitulate the observed effects of $P_i$ (*Figure 5g,h*).

## Discussion

The time resolution afforded by the UFFC system has allowed us to gain novel insights into dynamics of the cardiac myosin initial states following actin attachment and the working stroke. We resolved a short-lived state that is capable of bearing loads up to 4.5 pN and obtained the first estimates of the delay between myosin attachment to actin and the working stroke as a function of applied load from single molecule measurements. We also observed the effects of free phosphate on these events. The results allow us to characterize the order and properties of the rapid processes which follow cardiac myosin association with actin and lead to force generation. Based on the experimental results discussed below, we propose a model (*Figure 6*) in which an early actomyosin state that is short-lived but nevertheless is capable of bearing load leads promptly to the working stroke and concomitant strong binding, which is then followed by slower $P_i$ release.

### Short-lived states are Stereo-Specific, Open-cleft, Pre-stroke, and $P_i$-bound

Our UFFC experiments reveal short-lived interactions which detach before myosin completes its biochemical cycle via ATP binding. Although we have used a two-headed, HMM construct in our experiments to emulate biological structure, we did not observe changes in the stiffness (measured by the bead covariance signal) during attachments, or lifetime distributions consistent with paired dissociation events that might signal double-headed attachment. Thus, the data are interpreted as resulting from single myosin heads. The short-lived state detected in this study can stop the motion of an actin filament and maintain forces of up to 4.5 pN for hundreds of microseconds. In earlier publications, a weak-binding state of myosin has been described in several ways, and we considered whether the observed short-lived interactions may be in such a weak-binding state.

An electrostatic, non-stereospecific interaction between actin and myosin has been called a weak-binding state in which the myosin head experiences significant orientation fluctuations relative to actin. (*Thomas et al., 1995*; *Thomas et al., 2002*; *Ostap et al., 1995*; *Bershitsky et al., 1997*; *Ferenczi et al., 2005*; *Wu et al., 2010*; *Kraft et al., 2005*; *Lowy et al., 1991*; *Cooke et al., 1982*) The dissociation rate of the short-lived state found here ($k_f$) is substantially more force dependent under hindering loads as compared to assisting loads (*Figure 2f*, *Figure 2—figure supplement 2*) and as mentioned above it can briefly support loads. These characteristics suggest that the actomyosin bond in this state is stereo-specific and not a highly disordered interaction.

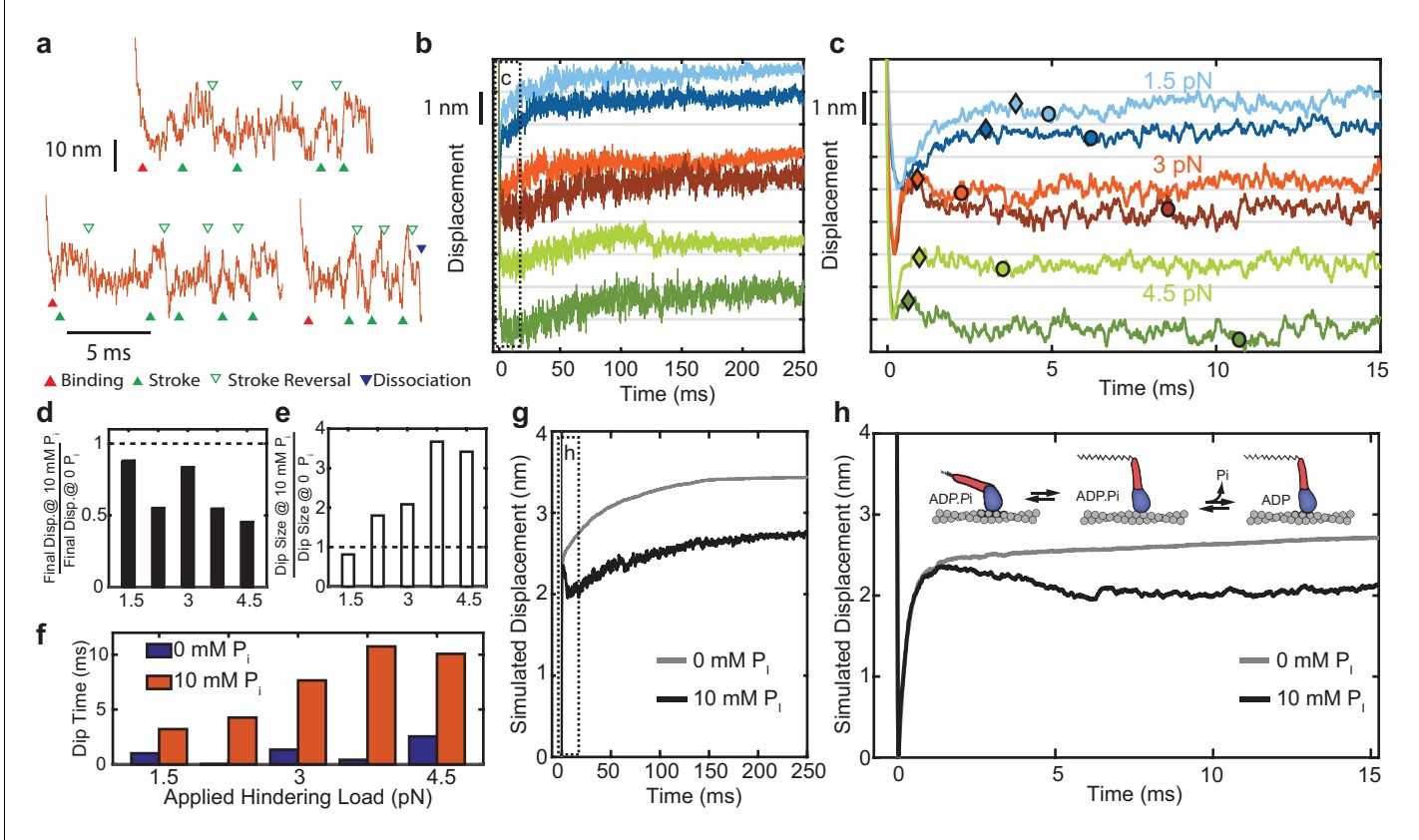

**Figure 5.** Effect of added $P_i$ on reversals of the working stroke. (**a**) Example traces of displacements in individual interactions under 3 pN of load at 10 mM $P_i$ with apparent strokes (filled green triangles) and reversals (open triangles) identified. (**b**) Ensemble averages of events longer than 25 ms at 1.5 pN (blue), 3 pN (red), and 4.5 pN (green) with 0 (lighter colored traces) and 10 mM $P_i$ (dark traces). Traces at the various forces are offset by 2 nm for visualization. (**c**) Expanded view of the boxed region in b) showing the initial stroke and later 'dip' attributed to stroke reversals. Triangles indicate the detected maximum initial displacement, and circles indicate the location and timing of the maximum 'dip' in the signal using a moving average, as described in Materials and methods. (**d**) Total displacement for events longer than 25 ms at 10 mM $P_i$ relative to that with no added $P_i$, ({displacement at 10 mM $P_i$} / {displacement at 0 $P_i$}). (**e**) Relative amplitude of the dip for 10 mM vs. 0 added $P_i$. (**f**) Comparison of the time between the maximum initial displacement and minimum of the dip for no added $P_i$ (blue) and 10 mM added $P_i$ (red). (**g**) Simulated ensembles averages from the model described in the text and Materials and methods with $P_i$ release occurring after the working stroke. (**h**) Expanded view of the boxed region in g) showing the dip in the displacement predicted by this model, similar to the experimental data. Inset shows a simplified view of the working-stroke-first model used in the simulations.

DOI: https://doi.org/10.7554/eLife.49266.015

The following source data and figure supplements are available for figure 5:

**Source data 1.** Matlab figure files with data from *Figure 5*.
DOI: https://doi.org/10.7554/eLife.49266.019

**Figure supplement 1.** Simulations of ensembles averages with and without added $P_i$.
DOI: https://doi.org/10.7554/eLife.49266.016

**Figure supplement 2.** Simulations of the initial ensemble displacement for immediate (blue) vs. multistep processes (red, yellow, green) occurring between actin binding and the working stroke compared to the observed data at 3 pN load (black).
DOI: https://doi.org/10.7554/eLife.49266.017

**Figure supplement 3.** UFFC Real Time Drift and EOD Slope Correction (DSC) system.
DOI: https://doi.org/10.7554/eLife.49266.018

Weak-binding has also been used to describe a conformation of the myosin head in which a cleft in the actin binding interface is open and is usually associated with ATP or ADP-$P_i$ bound myosin (*Sweeney and Houdusse, 2010*; *Dominguez et al., 1998*). This cleft-open state has been proposed to be the initial stereospecific interaction between actin and myosin which can rapidly dissociate from actin unless a relatively slow isomerization leads to 'strong binding' (*Thomas et al., 2002*;

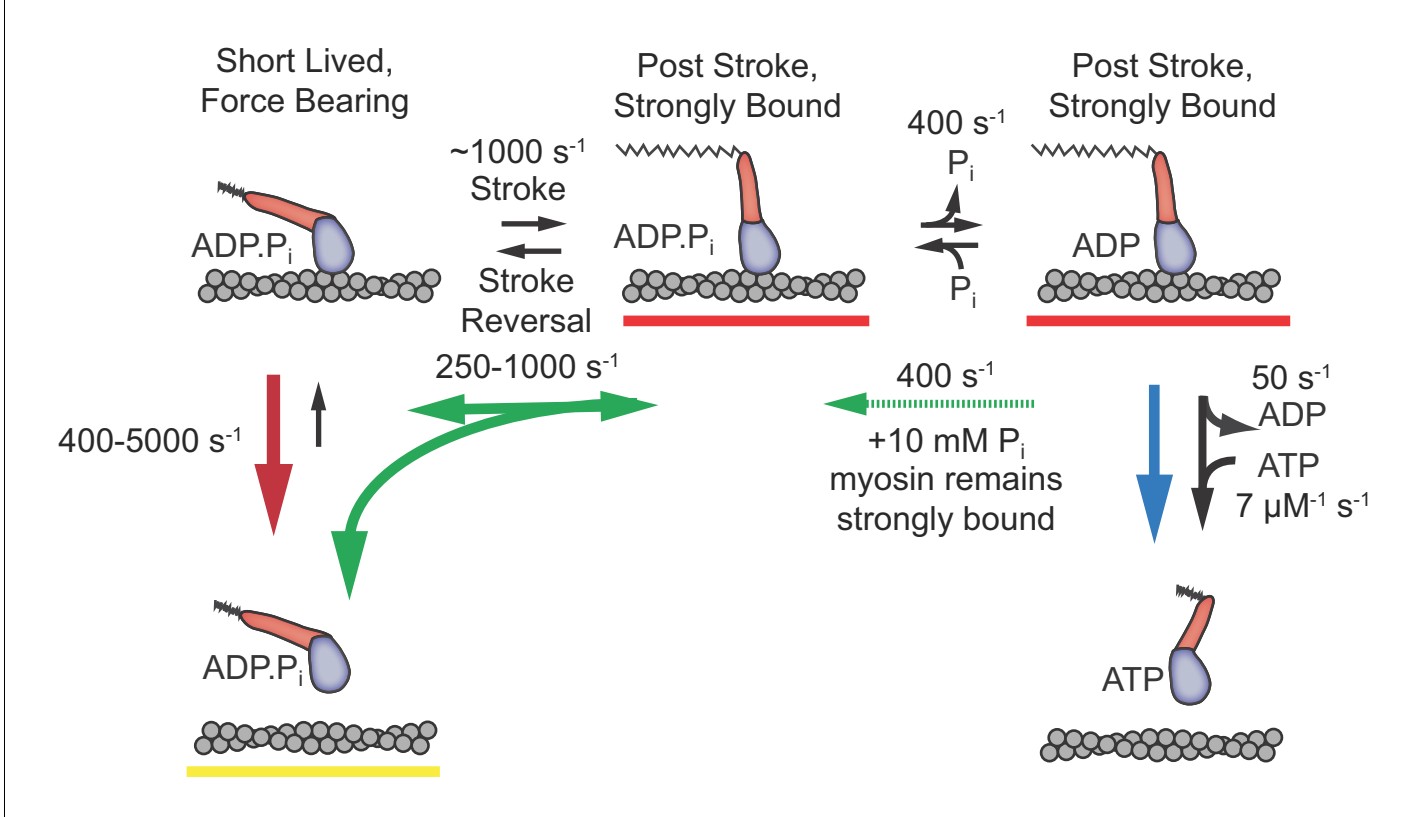

**Figure 6.** Proposed model with short-lived, weakly-bound attachments directly preceding the working stroke, which is followed by phosphate release and potential phosphate rebinding. Rates constants are estimates at zero load (lowest values) to 4.5 pN of hindering load (highest values). Colored arrows indicate processes associated with detachment rate $k_f$ (red), $k_{int}$ (green), and $k_s$ (blue). Dashed green arrow is only relevant when 10 mM $P_i$ is added. Red underlined states indicate myosin is strongly bound, while yellow underlines indicate a weakly bound or unbound state.
DOI: https://doi.org/10.7554/eLife.49266.020

*Bershitsky et al., 1997*; *Ferenczi et al., 2005*; *Kraft et al., 2005*; *Brenner et al., 2005*; *Beausang et al., 2013*; *Brenner et al., 1982*). The short-lived states we observed are estimated to dissociate at ~500 s$^{-1}$ in unloaded conditions, fairly close to the measured ~950–1450 s$^{-1}$ dissociation rate of actomyosin after actin cleft opening induced by ATP binding (*Liu et al., 2015*; *Deacon et al., 2012*; *Bloemink et al., 2014*). Comparing this to the <0.1 s$^{-1}$ dissociation rates measured for known cleft-closed states (*Woody et al., 2018a*), our short-lived state's lifetime suggests that myosin is in an open-cleft conformation.

Actin-activated $P_i$ release is dependent on a stereo-specific interaction between actin and myosin. If $P_i$ were released prior to the observed short-lived state, there would have to exist an additional stereo-specific state that is too rapid to observe. Based on the estimated rate of dissociation from an open-cleft state of 950 s$^{-1}$ given above, such a state is highly unlikely. Thus, our short-lived state is likely to be a stereospecific, actin-cleft-open state with $P_i$ bound.

### The stroke and cleft closure occur before $P_i$ is released

The model in *Figure 6* which places the working stroke before $P_i$ release is supported by the rapid observed rate of the stroke ($k_{stroke}$) and the similarity between $k_{stroke}$ and $k_f$ (*Figure 4*), as well as the observations of increased and delayed reversals in the presence of free $P_i$ (*Figure 5*), as discussed further below.

Ensemble averages of actin displacement during actomyosin events show that there is no appreciable working stroke during the short-lived interactions when time resolution is optimal (forces > 1.5 pN, *Figure 3d*). The working stroke appears to immediately follow this short-lived state, as the observed rate of the working stroke ($k_{stroke}$, diamonds in *Figure 4c*) is greater than or equal to the

rate of detachment from the short-lived state ($k_f$, circles in *Figure 4c*). $k_{stroke}$, surprisingly, shows an increased rate with larger forces in the direction that should hinder the working stroke. If the stroke immediately follows the short-lived state, this result can be readily explained by the slip-bond behavior of $k_f$, as the observed $k_{stroke}$ ($\approx k_{+5} + k_{-4}$, *Figure 1*) should increase when $k_{-4}$ increases. If a significant intermediate process, such as $P_i$ release, occurred between entering the short-lived state and the stroke, it would have to occur at >10,000 s$^{-1}$ to obtain the observed value of $k_{stroke}$ (*Figure 5—figure supplement 2*). In addition, if $P_i$ release and rebinding were to occur before the stroke, $k_{stroke}$ would be slowed by $P_i$ rebinding when it is present in solution (*Figure 4—figure supplement 1*). The lack of effect from 10 mM $P_i$ on $k_{stroke}$ (*Figure 4c*, compare open and closed diamonds) is thus further evidence that $P_i$ release and rebinding do not occur before the working stroke.

Biochemical estimates of the $P_i$ release rate for cardiac myosin in solution range from 11.6 to 17 s$^{-1}$, (*Rohde et al., 2017*; *Liu et al., 2015*) which is much slower than the observed stroke rate. The biochemical measurements estimate the microscopic rate of $P_i$ release, while our measurements are observed rates including competing processes (such as detachment) from the same states. If $P_i$ release were to happen before, or simultaneously with, the stroke at a rate of 17 s$^{-1}$, less than 0.4–4% (high load to zero load) of the molecules occupying the short-lived state would produce a stroke, the rest detaching at $\geq$500–5000 s$^{-1}$. The event duration data show that >10–40% (high - low load) of interactions proceed through the entire biochemical cycle (*Figure 2g*, *Figure 2—figure supplement 1*). This comparison thus again supports models in which $P_i$ is released after the working stroke and prevents reversal of the stroke and cleft re-opening unless $P_i$ rebinds.

## The stroke can be reversed and is reversed more often with $P_i$ present

It has been previously suggested that the myosin actin-binding cleft closure, and a corresponding increase in actin affinity, is tightly coupled to the working stroke (*Sweeney and Houdusse, 2010*; *Málnási-Csizmadia and Kovács, 2010*; *Xu and Root, 2000*), and our data support this hypothesis. If the binding interface cleft remained open after the stroke, a significant fraction of short-lived events would be expected to show a displacement of actin. Under conditions when the time resolution was optimal (forces $\geq$ 3 pN), negligible displacement was observed in short-lived events.

The data provide evidence that the working stroke and cleft closure are both reversible. The $k_{int}$ detachment rate without added $P_i$ showed an asymmetric force dependence about zero load (*Figure 2f*, green, *Figure 2—figure supplement 2*), consistent with a force-dependent rate for the reversal of the working stroke followed immediately by detachment from the short-lived, pre-stroke state. Evidence of these reversals appeared in the ensemble averages of displacement both with and without free $P_i$ (*Figure 3c,e* and *Figure 5b,c*) as a dip in the displacement that occurred within a few milliseconds after the initial stroke.

An increase in the fraction of rapidly detaching events was observed with 10 mM added $P_i$ at low forces (*Figure 2g,h*). This result is consistent with previous single molecule observations (*Baker et al., 2002*; *Debold et al., 2013*), and with the scheme of *Figure 6* in which $P_i$ rebinding allows more stroke reversals which can lead to premature detachments. At higher forces, it is likely that the increased fraction of direct detachments from the short-lived state (at rate $k_f$) may mask the effects of $P_i$ on attachment durations. Ensemble averages of the long events (>25 ms) show that added $P_i$ leads to a larger and more delayed dip in the displacement (*Figure 5c-f*), consistent with the scheme of *Figure 6* in which $P_i$ rebinding allows reversals to occur both more often and later in the attachment (*Figure 4—figure supplement 1*). The decreased displacement in the presence of 10 mM $P_i$ compared to control (*Figure 5d*) is consistent with myosin detaching from a pre-stroke conformation after a delayed reversal of the working stroke.

It should be noted that for this model to describe the ensemble average data (*Figure 5g,h*), the lifetime of the pre-stroke state following a reversal was approximately 1 ms under load (*Figure 5—figure supplement 1*). This is longer than the estimated 300 μs lifetime of the initial pre-stroke state under load. This might suggest that after $P_i$ is released, the $P_i$ rebinding does not cause the actin cleft to reopen fully, even if it does allow power stroke reversal. Additional studies and modeling will be required to test this speculation, but it has been suggested previously that $P_i$ rebinding may induce transitions off of the canonical pathway (*Debold et al., 2013*).

The model in *Figure 6* can explain all our observations. In this model, electrostatically mediated interactions between actin and myosin transition to the short-lived, stereo-specific state detected in our study. In this state, myosin can briefly bear force with an open actin binding cleft and $P_i$ bound.

The working stroke of myosin occurs directly from this state and the closure of the cleft is tightly coupled to the stroke. Both the stroke and cleft-closure can be reversed under zero load, with a rate ($k_{int}$) that increases with hindering load. Phosphate release occurs after the stroke and rebinding of phosphate allows reversals of the stroke.

### Relation to previous work

The scheme of *Figure 6* is fully consistent with a previously proposed model developed from muscle fiber experiments for skeletal myosin II (*Dantzig et al., 1992*; *Walker et al., 1992*). That model included a force bearing state followed by reversible force generation (the working stroke) and then phosphate release (*Dantzig et al., 1992*). Similar studies on cardiac fibers showed this model also is applicable to cardiac myosin (*Araujo and Walker, 1996*). More recent studies of the rate of the working stroke in skeletal (*Muretta et al., 2015*) and cardiac (*Rohde et al., 2017*) myosin II, and myosin V (*Trivedi et al., 2015*) using FRET probes and time-resolved fluorescence have found evidence of a conformational change in the motor domain, likely a rotation of the lever arm, that precedes phosphate release. The data from previous experiments on fast-skeletal myosin with the UFFC system support that the model of *Figure 6* also applies to this myosin isoform (*Capitanio et al., 2012*). The present data provide the most direct evidence for this ordering of events after actin attachment in cardiac muscle myosin.

High resolution crystal structures of myosin have been used to suggest that phosphate must be released before the stroke occurs (*Llinas et al., 2015*; *Sweeney and Houdusse, 2010*). In addition, some models of contraction developed to explain a wide variety of muscle fiber data have suggested that phosphate release and the working stroke are not tightly coupled (*Caremani et al., 2015*; *Caremani et al., 2013*). While our data cannot exclude that $P_i$ may occasionally be released before the stroke, if this occurred more than 20% of the time, there would be a discernable delay in the stroke rate in the presence of $P_i$ which was not observed (*Figure 4*). The possibility remains that other myosin isoforms which exhibit different kinetic properties than muscle myosin, such as non-muscle myosins V and VI, may proceed through actin binding, the working stroke, and phosphate release with different dynamics or even different sequences (*Llinas et al., 2015*; *Sweeney and Houdusse, 2010*; *Rosenfeld and Lee Sweeney, 2004*). Specific experiments on those systems are needed to clarify the mechanisms.

## Materials and methods

### Protein purification and source

A heavy meromyosin (HMM) construct of human β-cardiac myosin (MHY7) was expressed in C2C12 myoblasts and purified as previously described (*Winkelmann et al., 2015*). The HMM protein has 1146 residues that include residues 1–1138 of the human MYH7 gene and a FLAG tag on the C-terminus (res. 1139–1146). The sequence of the HMM preparations used in this study were confirmed by LC/MS/MS of protein digests. Bound light chains are those that are constitutively expressed in the C2C12 cells (MLC1/MLC3 and rLC2).

### Experimental protocol

Experiments were conducted in flow cells constructed and prepared as previously described (*Woody et al., 2018a*). The assay solutions contained 60 mM MOPs, 10 mM DTT, 1 mM EGTA, 1 mM MgCl$_2$, and 1 mg/mL BSA. The calculated ionic strength of the experimental solutions was held constant across ATP and $P_i$ conditions. The values of the components and total ionic strength of the solutions for the various $P_i$ and MgATP conditions are given in *Supplementary file 1* Table 3.

Myosin was added to nitrocellulose-coated chambers and allowed to attach nonspecifically to 2.5 μm silica pedestal beads on the surface and then the chambers were blocked with 1 mg/mL BSA as previously described (*Woody et al., 2018a*). The concentration of myosin (~0.2 ng/mL)) led to about 1 out of 10 bead locations showing interactions between myosin and actin. The final assay solution contained pM concentrations of 10–15% biotinylated actin (Cytoskeleton Inc,) from rabbit skeletal muscle stabilized by rhodamine phalloidin, followed by ~3 microliters of the final assay solution containing 5 ng mL$^{-1}$ 500 nm diameter beads (PolySciences Inc,), coated nonspecifically with neutravidin (*Woody et al., 2018a*) (ThermoFisher Inc).

Experiments were performed with trap stiffness 0.06–0.08 pN/nm and 4–5.5 pN pretension applied to the actin dumbbells. When a myosin molecule was found on a pedestal bead, a stage stabilization system was engaged (*Woody et al., 2018b*; *Capitanio et al., 2005*) and actomyosin events were recorded without force-clamp feedback to determine the size of the myosin working stroke and directionality of the filament (*Woody et al., 2018a*). The ultra-fast force clamp was then engaged, and 5–10 min of data were recorded for each applied force. Occasional filament breakage prevented collection at all forces (1.5–4.5 pN) for some dumbbells.

## Optical trapping implementation

The optical trapping setup used in these experiments was described previously in detail, (*Woody et al., 2017*) including optimizations to increase the update response time of the feedback loop to less than 8 μs. Briefly, polarization-split 1064 nm beams are steered by two 1D electro-optical deflectors (EODs, Conoptics, Inc) into a 60x water immersion objective (Nikon). The laser light collected from the chamber of an oil immersion condenser is projected onto two quadrant photodiodes conjugate to the back focal plane of the condenser for direct force detection (*Gittes and Schmidt, 1998*). The feedback loop and data acquisition used a National Instruments Field Programmable Gate Array controlled by custom LabVIEW virtual instruments.

Experiments were conducted otherwise as previously described (*Capitanio et al., 2012*; *Woody et al., 2018b*; *Gardini et al., 2018*) with applied forces of 1.5–4.5 pN in directions alternating toward each end of the actin filament. The trap moved under the applied force for 200 nm before the applied force was switched to the opposite direction and the cycle of motions back and forth repeated. To eliminate effects that tension on the actin filament might have on myosin interaction, the forces were distributed between the two beads so that the pretension on the filament was constant even as the applied force changed magnitude (*Figure 5—figure supplement 3*). The force signals and feedback output signals to the EODs were digitized at 250 kHz through a 125 kHz anti-aliasing filter in the sum and difference amplifiers for the *x*- and *y*-direction difference signals from the photodiode currents. The positions of the traps were determined from the EOD driving signals and optical calibration measurements of corresponding bead displacements.

We added a novel real-time drift and slope correction (DSC) system to our UFFC instrument to improve stability and accuracy of the force signal as a result of thermal changes and other variations. This system is described in the below and *Figure 5—figure supplement 3*.

## Real-time drift and slope correction (DSC)

Relative to acousto-optic deflectors (AODs) more commonly used in dynamic optical trap instruments, the electro-optic deflectors (EODs) used here provide a much smoother angular deflection of the IR beams (constant slope of deflection vs. input voltage) at the expense of maximum deflection angle (*Woody et al., 2018b*; *Valentine et al., 2008*). The EOD crystals are more narrow (2 x 2 mm) and much longer (100 mm) than typical AODs (~4 x 4 x 10 mm) and it is essential to keep the laser beams wholly within the crystals, so alignment is more critical and was checked and/or realigned daily.

Even with regular alignment, thermal drifts and or air currents produced instrumental 0.25–2 pN fluctuations over an hour period of recording. To obtain accurate and reliable force signals, we utilized an real time drift and slope correction (DSC) system which periodically pauses the feedback system during an experiment to probe the true 'zero load' ($F_0$) signal values and correct for the drift. While the feedback system is briefly paused, the trap positions are not updated and (when myosin is not interacting with actin) the only force on the beads is from the pretension applied to the filament. The force signals observed during each pause can be used to reset the zero-force voltage and calculate the setpoint in the feedback loop to the recorded $F_0$ level plus the applied force.

We implemented this system on the FPGA feedback controller such that every 10 ms the feedback loop pauses twice, first when the trap position reaches the top of the set excursion distance, and again when the trap position is at the bottom of the excursion (*Figure 5—figure supplement 3c*). Each pause lasts 2 ms, and the force signals from the last 1.5 ms of the pause are averaged for each bead to provide an estimate of the $F_0$ signals. These individual values are averaged over 20 pauses to calculate the setpoint during the experiment. Either the top or bottom pause allows correction for the slow thermal drifts. Additionally, by using both the top and bottom $F_0$ values and the

known excursion distance, the value of the EOD slope is be calculated (*Figure 5—figure supplement 3*).

$$EOD\ slope = \frac{F_{0_{Hg}} - F\_0_{Lw}}{d} \tag{1}$$

This slope value can then be used in the FPGA to update the setpoint value as the trap positions change in order to account for the slope in the force signals.

$$SetPoint = F_{set} + F_{pre} + F_0(x) \tag{2}$$

$$F_0(x) = (EOD\ slope)x + F_{0_{Lw}} \tag{3}$$

Examples of the drift offset and the calculated slope values over a 30 s trace are shown in *Figure 5—figure supplement 3d*.

When the data are analyzed, the recorded force signals can be transformed into the actual force signals by using a similar algorithm in the analysis software that was used in real time on the FPGA (*Figure 5—figure supplement 3e*). When detecting and characterizing binding events, the forces and positions of the traps when the feedback was paused are ignored. This DSC system markedly improved reliability and consistency within and across experiments, reduced the width of the velocity distributions, lowered effective deadtimes, and improved data reliability at lower forces.

## Data analysis

### Event detection

Events were detected using a modified version of software written in Matlab (Mathworks, Natick, MA) used previously for UFFC experiments (*Capitanio et al., 2012*; *Gardini et al., 2018*). Briefly, the velocity of the traps is calculated and smoothed using a 160–800 μs width Gaussian filter, and a threshold is determined for distinguishing periods when the trap motion has stopped that balances false binding and false unbinding events. The amount of smoothing is set so that the number of false events is estimated to be <1% of all detected events based on statistical estimates (*Colquhoun and Sigworth, 1995*). The smoothing varies with the applied force but is constant across experimental conditions. Because higher loads cause faster motion of the actin, less smoothing is required and shorter events can be detected at higher loads. Small corrections are applied to the start- and end-times of the events to account for delay based on the Gaussian filter, as previously described (*Capitanio et al., 2012*). The estimated uncertainty of the localization of binding time ($\sigma_t$) can be calculated and ranged from 500 μs at 1.5 pN to <60 μs at 4.5 pN applied load. From each data trace, a theoretical minimum detectable event length (deadtime) was calculated based on the velocity distributions and the threshold value as previously described (*Capitanio et al., 2012*). Events were designated as occurring either during hindering or assisting load based on the direction of actin motion when binding occurred and the polarity of the actin as determined by initial non-feedback experiments.

### Duration analysis

For a given force, all events which were shorter than the largest calculated deadtime for traces recorded at that force were excluded from the analysis. The deadtimes, set to be constant across different biochemical conditions, were 2.7, 1.25, 0.73, 0.65, and 0.5 ms for forces 1.5, 2.25, 3, 3.75, and 4.5 pN respectively. Events from molecules at the same magnitudes and directions of force were pooled for each given condition (e.g. forces, concentrations of MgATP and free $P_i$). Parameters for deadtime corrected single, double, and triple exponential distributions were estimated in MEMLET (*Woody et al., 2016*) for each data set and the log-likelihood ratio test was used with a p-value cutoff of 0.05 to determine if the triple and/or double exponential distributions were statistically justified. Each molecule contributed equally to the fitted values as described previously (*Woody et al., 2018a*), although this did not lead to considerable differences compared to the fits weighted by the number of interactions. The observed amplitudes of each phase are reported as well as the deadtime corrected amplitudes (*Figure 2—figure supplement 1*), which consider how many events are

likely missed due to the experimental deadtime and assuming exponential duration components (*Woody et al., 2016*).

## Ensemble averaging

Ensemble averages were carried out as have been described previously (*Woody et al., 2018a*). The beginnings of the events were aligned based on their detected time of binding from the crossing of the smoothed velocity trace through the detection threshold as previously described (*Capitanio et al., 2012*). The trap position for the leading bead (the bead with the higher magnitude of force applied) was taken as the position of the actin filament, as there is less influence of non-linear end compliance on this more highly loaded bead. The extension point was taken to be the displacement at time $\sigma_t$ (uncertainty of binding time) before the detected end of the event, meaning this value was used in the averages for all time points longer than the actual event duration. The value of $\sigma_t$ depends on the smoothing of the velocity for event detection, and thus was smaller (more accurate) for larger forces. The average value of $\sigma_t$ varied from 0.5 ms for 1.5 pN to ~72 μs for 4.5 pN. Ensembles were weighted such that each molecule contributed equally to the final average.

Events longer than 15 ms were analyzed together for the 0 $P_i$ conditions in *Figure 3* to isolate events which went through the entire biochemical cycle, including ATP binding to terminate the actomyosin interaction, as the total observed step did not change >10% by excluding any additional events longer than 15 ms (*Figure 3—figure supplement 1a*). For the 10 mM $P_i$ data and for analysis of the 0 $P_i$ data which was directly compared to that with 10 mM $P_i$ (*Figures 4* and *5*), events longer than 25 ms were included because the total working stroke size continued to increase as events between 15 ms and 25 ms were excluded (*Figure 3—figure supplement 1b*), which could be due to reversals and detachments occurring later in an interaction when $P_i$ is present.

## Quantifications of ensemble averages

Quantification of the ensemble average was done programmatically, using the methods and parameters reported below. The minimum position of the ensemble average within the first millisecond of detected actin binding was set to be zero displacement. The size of the initial displacement was calculated from the raw (unfiltered) ensemble averages by finding the time ($t_{init}$) when the ensemble average reached its maximum within the first 1.2 ms following the minimum position, with the exception of the 1.5 pN data, which were searched over the first 5 ms. The displacement from a window of 200 μs centered on $t_{init}$ was averaged to determine the maximum initial position. The minimum and maximum positions were used to normalize the displacements to range from 0 to 1, as plotted in *Figure 4a,b*. The normalized displacement data between the minimum position and $t_{init}$ was used to fit a single exponential rise with an amplitude of 1 to determine the rate of the working stroke as reported in *Figure 4c*.

For quantification of the dip size and time, the initial stroke size was determined by averaging 100 μs on either side of $t_{init}$. The location and position of the dip were determined by taking a 1 ms moving average of the ensemble and finding the minimum point between $t_{init}$ and 15 ms after the detected binding time. The size and timing of the dip are reported as the difference between the time and displacement of initial stroke and the minimum dip respectively. The total displacement is quantified as an average over the last 200 μs of the ensemble average.

## Modeling

Simulations of individual traces were performed using a Monte-Carlo based method (*Gillespie, 1977*) using the rates and transitions given in *Figure 5—figure supplement 1*. For each simulated state ($i$), and for each transition out of that state ($i \rightarrow j$), times, $t_{i \rightarrow j}$, were randomly selected from an exponential distribution based on the transition rate ($k_{i \rightarrow j}$). The shortest $t_{i \rightarrow j}$ was used as the lifetime of the state $i$ and the transition associated with that time was used to assign the next state ($j$). This method leads to the expected results that the flux through a transition ($A_{i \rightarrow j}$) will be the rate of that transition $k_{i \rightarrow j}$ divided by the sum of all rates exiting the current state $\left( k_{sum} = \sum_j k_{i \rightarrow j} \right)$. The effective rate of the transition will be $k_{sum}$. Once all states were assigned for a given simulated interaction (the last state is always a detached state), a position value was assigned to each state based on the mechanical properties of the states. The simulated position values were drawn for each interaction from a

Gaussian distribution with a 0.1 nm standard deviation from their set mean value. The position trace before the initial interaction was a constant slope with a velocity similar to what is observed in the data for the simulated applied force. Upon initial binding, an exponential function was used to simulated stretching of the myosin under the applied load using a linear stiffness of 2 pN/nm. 3000 interactions were simulated, and the interactions lasting longer than a 25 ms cutoff were included for averaging with the same routine used for the data analysis.

## Additional information

### Funding

| Funder | Grant reference number | Author |
|---|---|---|
| National Institutes of Health | P01-GM087253 | E Michael Ostap<br>Yale E Goldman |
| National Institutes of Health | R01-HL133863 | Donald Winkelmann<br>E Michael Ostap |
| National Science Foundation | CMMI: 15-48571 | E Michael Ostap<br>Yale E Goldman |
| National Science Foundation | NSF GRFP | Michael Woody |
| National Institutes of Health | R35-GM118139 | Yale E Goldman |
| Horizon 2020 Framework Programme | 654148 | Marco Capitanio |
| Ente Cassa di Risparmio di Firenze | | Marco Capitanio |

The funders had no role in study design, data collection and interpretation, or the decision to submit the work for publication.

### Author contributions

Michael S Woody, Conceptualization, Data curation, Software, Formal analysis, Investigation, Visualization, Methodology, Writing—original draft, Writing—review and editing; Donald A Winkelmann, Resources, Funding acquisition, Writing—review and editing; Marco Capitanio, Software, Methodology, Writing—review and editing; E Michael Ostap, Conceptualization, Resources, Supervision, Funding acquisition, Visualization, Methodology, Writing—original draft, Project administration, Writing—review and editing; Yale E Goldman, Conceptualization, Supervision, Funding acquisition, Investigation, Methodology, Writing—original draft, Writing—review and editing

### Author ORCIDs

Michael S Woody (iD) https://orcid.org/0000-0002-8292-2695
Yale E Goldman (iD) https://orcid.org/0000-0002-2492-9194

### Decision letter and Author response
Decision letter https://doi.org/10.7554/eLife.49266.024
Author response https://doi.org/10.7554/eLife.49266.025

## Additional files

### Supplementary files
• Supplementary file 1. File containing three supplementary tables which provides n-values for experiments, measured stroke rates, and ionic strength conditions.
DOI: https://doi.org/10.7554/eLife.49266.021

• Transparent reporting form
DOI: https://doi.org/10.7554/eLife.49266.022

## Data availability

The primary optical trapping data files are > 1TB, and are available upon request. Source data files have been provided for Figures 1-5.

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
