## [Decision Letter]

Thank you for submitting your article "Single molecule mechanics resolves the earliest events in force generation by cardiac myosin" for consideration by *eLife*. Your article has been reviewed by three peer reviewers, one of whom is a member of our Board of Reviewing Editors, and the evaluation has been overseen by Olga Boudker as the Senior Editor. The following individuals involved in review of your submission have agreed to reveal their identity: Michael Geeves (Reviewer #2); Arne Gennerich (Reviewer #3).

The reviewers have discussed the reviews with one another and the Reviewing Editor has drafted this decision to help you prepare a revised submission.

The reviewers and I felt that this is a significant piece of work that needs to be widely read and understood. It is a major technical achievement – to have such high time resolution optical trap data and the authors have chosen an ideal protein to use. Rather than the fast myosins used in Captianio's earlier work laser trap studies they use a β cardiac myosin which is several fold slower for some of the key molecular events. This allows fine detail of myosin attachment and the subsequent events to be mapped in great detail

Here for the first time the authors appear to be seeing myosin heads weakly attached to actin and the transition through the power stroke to a strongly attached form, significantly with phosphate (P_i_) still present in the motor. Each of these is a significant observation and if the interpretations are correct then the results have a big impact on some of the major questions about how myosin generates force. The interpretation of the data will be argued over for some time.

Arguments have swung back and forth about the order of the power stroke and P_i_ release steps, which comes first? The inability to measure these events directly has led to intense debate. Here they come down very firmly on power stroke followed by P_i_ release.

Another important question is what is the nature of the weakly-attached pre power-stroke state and how much load can it bear? – Here the result is up to 5 pN but with a diminishing lifetime at higher load. Importantly the transition forward is less likely when attachment occurs at high load not because of load inhibition of the forwards step but because of acceleration of the reverse step.

Overall it was agreed that this work is important and that your conclusions are well supported by the data. In places the reviewers felt that the presentation needs to be improved by changes to the text.

Essential revisions:

1) The authors do not state clearly in the main text what the myosin construct is that they are using; human β HMM with a flag tag. The details are important and should not be left in the Materials and methods alone. It is only in the 2nd paragraph where they state that this is human β myosin but do not say this is HMM or how it is attached to the surface. The experiments would not be possible with any myosin faster than β and to my knowledge it is only the human β protein that is currently available in a form that can be expressed with tags.

In summary the reasons for using the β-cardiac myosin, rather than other isoforms, need to be explained clearly in the text.

2) The authors do provide a brief paragraph at the end of the manuscript referring to other myosins. For a general reader the possibility of isoform differences between myosins needs to be addressed in relation to the major conclusions.

3) The use of HMM rather than long S1 does raise issues of any involvement of the 2nd head in the events observed. This is not mentioned in the paper but will add a complication to interpretations of the data. The authors should address why HMM is used and how this affects their interpretation

4) Figure 2—figure supplement 2, in contrast to the data shown on a log time base in Figure 2 does show a marked asymmetry in the stiffness (or d value in their fits). This requires some comment and may be one of the very interesting features of the data that requires further explanation. In contrast Figure 2—figure supplement 2 is instead as showing symmetry, a single d value for assisting and hindering loads. There has been a lot of debate about asymmetry in the crossbridge stiffness and its consequences. This is worth commenting on.

5) Many of the issues here have been long debated and in this short report the authors cannot be expected to cover all the literature but some of the citations reflect a narrow view of the debate. For example:

a) In several places the authors state that their data is compatible with published data and then refer to the published data on pig myosin (subsection “Short lived States are Stereo-Specific, Open-cleft, Pre-stroke, Pi-bound states”, third paragraph, subsection “Duration of actomyosin interactions observed under load”, fifth paragraph). There is a substantial literature on the human protein including from some of the authors. The work on human form should surely be a better reference.

b) Similarly the references to the weakly attached sate (Bagshaw and Trentham, 1974; White, Belknap and Webb, 1997; Lymn and Taylor, 1971) seem an odd mixture. Bagshaw and Trentham, 1974, is all work in the absence of actin.

---

## [Author Response]

Essential revisions:1) The authors do not state clearly in the main text what the myosin construct is that they are using; human β HMM with a flag tag. The details are important and should not be left in the Materials and methods alone. It is only in the 2nd paragraph where they state that this is human β myosin but do not say this is HMM or how it is attached to the surface. The experiments would not be possible with any myosin faster than β and to my knowledge it is only the human β protein that is currently available in a form that can be expressed with tags.In summary the reasons for using the β-cardiac myosin, rather than other isoforms, need to be explained clearly in the text.

We have added to the main text to clearly state the nature of the construct:

“The myosin protein was a heavy meromyosin (HMM) construct with a Flag-tag placed at the N-terminus for purification. […] Although this is a two-headed construct, we did not observe evidence that both heads interacted simultaneously with actin in the presence of ATP (see Discussion).”

We also have added to the Discussion about the isoform in response to questions 2 and 3 as mentioned below.

2) The authors do provide a brief paragraph at the end of the manuscript referring to other myosins. For a general reader the possibility of isoform differences between myosins needs to be addressed in relation to the major conclusions.

We expounded upon the last paragraph to further emphasize that there may be differences between the various myosin isoforms.

“The possibility remains that other myosin isoforms which exhibit different kinetic properties than muscle myosin, such as non-muscle myosins V and VI, may proceed through actin binding, the working stroke, and phosphate release with different dynamics or even different sequences (Llinas et al., 2015; Sweeney and Houdusse, 2010; Rosenfeld and Sweeney, 2004). Specific experiments on those systems are needed to clarify the mechanisms.”

3) The use of HMM rather than long S1 does raise issues of any involvement of the 2nd head in the events observed. This is not mentioned in the paper but will add a complication to interpretations of the data. The authors should address why HMM is used and how this affects their interpretation

We have added the following sentences near the beginning of the Discussion:

“Although we have used a two-headed, HMM construct in our experiments to emulate biological structure, we did not observe changes in the stiffness (measured by the bead covariance signal) during attachments, or lifetime distributions consistent with paired dissociation events that might signal double-headed attachment. Thus, the data are interpreted as resulting from single myosin heads.”

4) Figure 2—figure supplement 2, in contrast to the data shown on a log time base in Figure 2 does show a marked asymmetry in the stiffness (or d value in their fits). This requires some comment and may be one of the very interesting features of the data that requires further explanation. In contrast Figure 2—figure supplement 2 is instead as showing symmetry, a single d value for assisting and hindering loads. There has been a lot of debate about asymmetry in the crossbridge stiffness and its consequences. This is worth commenting on.

The distance parameters (*d*) values obtained from the Bell fit have been used as an estimate of the force dependence of the kinetic transitions being observed. This asymmetry in force sensitivity may be related to myosin stiffness, but it does not have to be. We interpreted the difference in the *d* values for *k*_f_ for assisting vs. resisting loads to be due to the transition of pulling myosin forward off of actin vs. backwards off of actin to be fundamentally different in terms of the molecular interactions. For *k*_int_, we have interpreted the single *d* value for assisting and resisting loads to indicate that a single transition is affected for both directions of force which slows under assisting load. This directionality is consistent with reversals of the working stroke. These points are outlined in the subsection “The myosin working stroke is rapid and more likely for long events” and subsection “Experimental protocol”.

5) Many of this issues here have been long debated and in this short report the authors cannot be expected to cover all the literature but some of the citations reflect a narrow view of the debate. For example:a) In several places the authors state that their data is compatible with published data and then refer to the published data on pig myosin (subsection “Short lived States are Stereo-Specific, Open-cleft, Pre-stroke, Pi-bound states”, third paragraph, subsection “Duration of actomyosin interactions observed under load”, fifth paragraph). There is a substantial literature on the human protein including from some of the authors. The work on human form should surely be a better reference.

We thank the reviewer for pointing this out and have updated the references to cite the most relevant work done on the human protein, namely Deacon, et al., 2012, and Bloemink et al., 2014 in the places listed.

b) Similarly the references to the weakly attached sate (Bagshaw and Trentham, 1974; White, Belknap and Webb, 1997; Lymn and Taylor, 1971) seem an odd mixture. Bagshaw and Trentham, 1974, is all work in the absence of actin.

We thank the reviewer for pointing this out and have updated the references to include Stein, Chock, and Eisenberg, 1984 and Brenner 1991.